# Ultrafine-Grained Zn–Mg–Sr Alloy Synthesized by Mechanical Alloying and Spark Plasma Sintering

**DOI:** 10.3390/ma15238379

**Published:** 2022-11-24

**Authors:** David Nečas, Jiří Kubásek, Jan Pinc, Ivo Marek, Črtomir Donik, Irena Paulin, Dalibor Vojtěch

**Affiliations:** 1Department of Metals and Corrosion Engineering, Faculty of Chemical Technology, University of Chemistry and Technology, Prague Technická 5, 166 28 Prague, Czech Republic; 2Department of Functional Materials, Institute of Physics of the Czech Academy of Sciences, Na Slovance 1999/2, 182 21 Prague, Czech Republic; 3Department Physics and Chemistry of Materials, Institute of Metals and Technology, University of Ljubljana, Lepi pot 11, SI-1000 Ljubljana, Slovenia

**Keywords:** metals and alloys, biomaterials, mechanical alloying, sintering, nanostructure, compression test

## Abstract

Zinc materials are considered promising candidates for bioabsorbable medical devices used for the fixation of broken bones or stents. Materials for these applications must meet high mechanical property requirements. One of the ways to fulfil these demands is related to microstructure refinement, particularly the decrease in grain size. In the present work, we combine two powder metallurgy techniques (mechanical alloying—MA, and spark plasma sintering—SPS) to prepare Zn–1Mg–0.5Sr nanograin material. The microstructure of compacted material consisted of Zn grains and particles of Mg_2_Zn_11_ intermetallic phases from 100 to 500 nm in size, which resulted in high values of hardness and a compressive strength equal to 86 HV1 and 327 MPa, respectively. In this relation, the combination of the suggested techniques provides an innovative way to form extremely fine microstructures without significant coarsening during powder compaction at increased temperatures.

## 1. Introduction

Zinc is considered one of the most prospective materials used for biodegradable applications due to its excellent biocompatibility and reasonable corrosion rate without the formation of toxic corrosion products or the release of hydrogen. However, its poor mechanical properties (yield stress ~20 MPa, ductility ~12 % [1]) are insufficient for most medical applications. Improvements are generally achieved by suitable alloying and thermomechanical processing (extrusion, rolling, and drawing). During processing, the dynamic recrystallization leads to the grain refinement to sizes of several μm, causing material strengthening according to the Hall–Petch relation. However, it is difficult to further decrease the grain size of the zinc matrix to support higher yield strength due to the low recrystallization temperature of zinc and its alloys (≈0 °C) [2]. Other strengthening mechanisms such as solid solution strengthening and secondary phase strengthening can affect the material’s behavior, but the quantity of these contributions is limited on the basis of the material’s chemical composition [1,3,4]. Zn–Mg-based materials are widely considered as the perspective for bioabsorbable medical devices, but the solubility of Mg in Zn is limited to almost zero at ambient temperature, and the size of the Mg_2_Zn_11_ intermetallic phases reaches relatively high values between 1 and 10 μm, even for extruded materials. Therefore, it is highly challenging to affect these characteristics in the desired direction (lower grain and particle size, and higher Mg concentration in solid solution), supporting the improvement of mechanical properties.

To progress with Zn–Mg-based materials, mechanical alloying (MA) as the main processing technique is suggested. The MA is a method based on a combination of repetitive cold welding and braking of powder particles during milling with balls in the specific vessel. Diffusion processes during the material processing led to the formation of metastable phases like supersaturated solid solutions or a wide range of intermetallic phases [5]. The process can be affected by several factors such as the rotation speed of the vessel, milling time, temperature of the process, milled material, and size of the vessels and milling balls, etc. [6]. Based on these conditions, high-energy milling may produce homogeneous fine-grained alloy powders with enhanced solubility of alloying elements in a solid solution over a thermodynamically stable concentration [5,7]. MA was found to be one of the methods that is capable of forming nanocrystalline structures. Other techniques that allow grain refinement close to 0.5 µm are equal-channel angular pressing (ECAP), high-pressure torsion (HPT), and accumulative roll bonding [8]. To preserve the microstructure of mechanically alloyed powders, a fast compaction technique, such as spark plasma sintering (SPS), is suitable for subsequent consolidation [9]. The SPS is a compaction method utilizing a combination of pressure and heating by Joule’s heat, which is generated by a high direct current that comes through the compressed powder. The surface between powder particles has significant resistance that leads to an increase in temperature [10,11] of the sintered sample. This method generates compact samples with low porosity [6]. 

In the present work, we prepared the Zn–1Mg–0.5Sr (wt.%) alloy from powders of pure metals by mechanical alloying and subsequent compaction by SPS. The magnesium and strontium addition are considered due to the positive effect of these elements on mechanical properties and materials’ biocompatibility [12,13,14,15,16]. However, ternary Zn-Mg–Sr alloys have been successfully prepared by casting [12,13,14] and thermomechanical processing such as extrusion [14], hot rolling [12,13], or even ECAP [17]; none of these studies has brought materials with unique, extremely fine-grained microstructure enabling the improvement in materials performance.

## 2. Materials and Methods

### 2.1. Materials Synthesis

Zn–1Mg–0.5Sr alloy (wt.%, Table 1.) was prepared by mechanical alloying (MA) for 120 min at 800 rotation per minute (RPM) using a Retch E-max mill equipped with a cooling system that enabled the temperature to be kept below 50 °C. The ball-to-powder ratio was selected as 10:1. The pure metallic powders: zinc (99.9 %, particle size <149 µm), magnesium (99.8 %, particle size <44 µm), and strontium (99.8 %, particle size <500 µm) were selected as initial materials. The weight of the powder mixture was 30 g. To prevent agglomeration of the powder mixture during mechanical alloying, 0.08 g of stearic acid was added to the powder mixture as a process control agent (PCA). The grinding vessels were rinsed with Ar 99.96 % atmosphere against oxidation during MA. The prepared powder mixture and pure Zn powder were compacted using SPS (FCT System HP-D 10) at 300 °C with a heating speed of 10 °C/s, 19.1 MPa for 10 min in a graphite die under a protective Ar atmosphere of 99.96 %. The temperature was selected according to our experiences with other Zn-based alloys. Lower temperatures caused an increase in materials porosity, higher temperatures support the coarsening of the microstructure and possible melting at particle boundaries, causing the cracking of graphite vessels. The development of conditions during the SPS method is shown in Figure 1. Material prepared by SPS was partially dissolved in a 20 % solution of HNO_3,_ and solutions were analyzed by Atomic absorption spectroscopy (AAS, 280 FS AA SPECTROMETER, Agilent, Santa Clara, CA, USA).

### 2.2. Microstructure Analyses

The samples for the microstructural observations were ground using SiC papers (up to P2500) and polished using diamond paste D2 (2 µm) and Eposil F suspension (0.1 µm, Al_2_O_3_). The microstructure of the prepared samples was characterized using an optical microscope (Eclipse MA200, Nikon, Minato, Japan) and a scanning electron microscope (SEM—VEGA 3 LMU, tungsten filament, Tescan, Brno, Czech Republic) with an Energy dispersive X-ray spectroscopy (EDS—Aztec, Oxford Instruments, Tubney Wood, UK). Material porosity was measured using Xradia 610 Versa (µCT) (ZEISS, Oberkochen, Germany) under the following conditions: resolution 7 µm; 160 kV, HE3 filter, and a detector distance of 150 mm. These measurements were performed using cuboid samples with 3 × 3 × 3 mm^3^ dimensions cut from the center and the edge of the sintered billet. In addition, smaller areas of the cubes were measured using a higher resolution (1 µm) to reveal the presence and character of smaller pores in the material structure. The results were processed by histotrophic segmentation (pixel intensity) using Dragonfly software (version 2022.1.B.1249, Object Research Systems, Montréal, Canada). Finally, the phase composition was measured by X-ray diffraction (X’Pert3 Powder instrument in Bragg–Brentano geometry using a Cu anode with scanning speed 0.055974°/s, Malvern Panalytical, Malvern, UK). The names and reference codes of the evaluated phases in the PDF 4 database were Zn (01-078-9363), Mg_2_Zn_11_ (04-007-1412), and SrZn_13_ (04-013-4885). The details of the microstructure were studied by transmission electron microscope (TEM—EFTEM Jeol 2200 FS, accelerating voltage 300 kV, LaB6, Jeol GmbH, Tokio, Japan). Firstly, thin strips (thickness <100 µm) were prepared by grinding and further thinned by Gatan’s PIPS polishing system by Ar ions (Gatan, Pleasanton, CA, USA). The average particle, grain, and intermetallic phase size were measured on SEM and TEM images and determined based on the measurements of Feret’s diameters. In such cases, the distances between the two parallel lines in horizontal or vertical directions restricting the object from its edges were measured and averaged for each particle, grain, or intermetallic phase.

### 2.3. Mechanical Properties Testing

The mechanical properties of the alloy were characterized by Vickers hardness and compression measurements. The HV1 was measured on a Future-Tech FM-100 at a load of 1 kg. A minimum of ten indentations were performed, from which the average and deviation were calculated. Compression tests were performed on three specimens with size 3 × 3 × 3 mm^3^ at a strain rate equal to 0.001 s^−1^ (Instron 5882) at ambient temperature.

## 3. Results and Discussion

### 3.1. Microstructure

The Zn–1Mg–0.5Sr powder prepared by mechanical alloying and the original Zn powder (Figure 2) were characterized by particle size from 25 to 140 μm (average size 70 μm) and 80 to 150 μm (average size 115 μm), respectively. Particles prepared by MA (Figure 2B) are of various irregular shapes with significant surface morphology and sharp edges as a consequence of the repetitive breaking and welding of powders and intensive plastic deformation compared to elongated particles with a relatively fine surface for pure zinc. Both powders were subsequently compacted by the SPS method. The microstructures of Zn–1Mg–0.5Sr alloy and Zn are shown in Figure 3. Both materials are composed of slightly deformed particles of original powder, separated by oxide shells (dark interface, white arrows in Figure 3, Figure 4 and Figure 5). Similar microstructure features were observed for other Zn-based materials [18] prepared by SPS. A detail of the microstructure of Zn–1Mg–0.5Sr alloy is shown in Figure 4. Material contained locally tightly arranged alternating layers of Zn matrix and intermediate phases (Mg_2_Zn_11_, oxides), which were formed due to the intensive plastic deformation during mechanical alloying. Overall, the Zn–1Mg–0.5Sr is characterized by an extremely fine microstructure. To clarify this significant modification, the microstructure of as-casted Zn–1Mg–0.5Sr is shown in the Appendix A. 

The grain size of Zn–1Mg–0.5Sr ranged from 100 to 500 nm (Figure 6B), with an average value of 193 nm. For comparison, the sample of compacted Zn achieved an average grain size of 2.7 µm (Figure 6B). Therefore, fast processing by SPS prevented coarsening of the microstructure during compaction, although this is not generally observed for the processing of zinc at increased temperatures (sintering, hot extrusion, hot isostatic pressing, etc.). The darker particles (red arrows in Figure 4A) correspond to the Mg_2_Zn_11_ intermetallic phases ranging in size from tens of nm to ≈700 nm, with an average particle size of 459 nm. The X-ray diffraction (Figure 7) showed that the alloy contained 5.2 wt.% of Mg_2_Zn_11_ and 4.0 wt.% of SrZn_13_. On the basis of the distribution of elements in the microstructure (Figure 4B), it is suggested that Sr is enriched, especially at grain boundaries, where it may exist in the form of fine ZnSr_13_ or segregation. In addition, several holes observed in the microstructure were enriched by Sr according to scanning electron microscope–Energy Dispersive X-ray Spectroscopy analyses (SEM–EDS analyses), indicating that SrZn_13_ particles were partially dissolved during sample preparation for microstructure analyses due to their high susceptibility to etching. Magnesium is predominantly contained in the intermediate phases, whereas 0.4 wt.% is covered by Mg_2_Zn_11,_ and 0.5 wt.% remains for oxides in the microstructure. These oxides are presented in the microstructure-like fine particles distributed mainly at the grain boundaries (Figure 5). To prove the origin of these particles, detailed EDS analyses using TEM were performed. The composition related to the light areas described in Figure 5b as oxides contained 35 ± 5 wt. % O, 20 ± 6 wt. % Mg, 45 ± 11 wt. % Zn. This indicates that these particles are oxides and contain rather a mixture of ZnO and MgO. Only 0.1–0.2 wt % of Mg was detected in the zinc solid solution, which is related to the low solubility of Mg in Zn at laboratory temperature. One can suggest that the solubility of Mg in Zn should be increased to metastable values during mechanical alloying, and indeed, we have observed that the zinc in the powder may obtain up to 0.6 wt.% of Mg in the form of solid solution. However, such conditions are not thermodynamically stable, and Mg has a very high tendency to form an intermetallic phase like Mg_2_Zn_11_ with a standard Gibbs energy of formation equal to ≈−235 kJ (at 298.15 K). Additionally, the standard Gibbs energy of the formation of one mole of MgO is even more negative (≈−570 kJ at 298.15 K). Therefore, MgO is precipitated in the microstructure, especially in areas with residual oxygen or oxygen access through diffusion at grain or particle boundaries. Eventually, Mg content in the solid solution is decreased close to the thermodynamically stable values (0.1 wt.%) in compacted samples.

Due to the various shape of milled powder particles and relatively low operating pressure during the SPS process, the porosity of Zn–1Mg–0.5Sr was determined by µCT. The results are shown in Figure 8. In the case of the sample from the center of the billet, the porosity reached 0.05 vol.%, and the pores were presented in two different arrangements—larger single pores and smaller pores interconnected by thin cracks. The larger individual pores reached the size of 20–100 µm, and the content of these pores in the center of the sample was relatively low. On the contrary, most pores were clustered into areas with mutual interconnection between individual pores. These areas reached almost 700 µm in size, and the maximal pore size in these areas was 50 µm. However, the average size of pores in these areas was approximately 20 µm. The relation between the position of the clusters and their size was not observed. The only difference between samples cut from the center and the edge of the sample was the pore content, which was almost twice as high (0.11 vol.%) at the edge of the billet. This increase was caused predominantly by larger clusters, which is documented in Figure 8 (red-colored pores).

In summary, the microstructure of Zn–1Mg–0.5Sr prepared by powder metallurgy techniques is unique due to the extremely low grain size and the existence of both very fine intermetallic phases and oxides. Almost chemically similar materials prepared by conventional methods reached the grain sizes of 10–50 µm for casted Zn–1Mg–1Sr [12]) and 2–10 µm for extruded Zn–0.8Mg–0.2Sr [14]. Furthermore, these materials were characterized by significant differences in microstructural aspects like the size of intermetallic phases and their arrangements, causing complications with anisotropy of mechanical properties. In the works of Ali et al. [19], Guleryuz et al. [20], and Yan et al. [21] binary Zn–Mg materials were prepared by a combination of MA (250–350 RPM, 4–8 h [19,20,21]) and hot sintering (350 °C, 4 h, 300 MPa [19]; 410 °C, 30 min, 30 MPa [20], 430–580 °C, 4 h + hot forging at 400 °C [21]). Prepared alloys reached much larger grain sizes (>80 µm) than in the presented case and contained structural defects such as high internal porosity, microcracks, and high volume fraction of oxides [20]. Some defects were partly eliminated in materials produced by compression-assisted techniques [19,21]. Nevertheless, the microstructures contained local segregations of pure magnesium, which were caused by insufficient energy generated by planetary mills during MA. Due to this, not all alloying elements were converted into intermetallic phases or solid solutions. These inhomogeneities significantly affect mechanical, corrosion, and other structurally sensitive properties. Yang et al. [22] focused on the preparation of mechanically alloyed Zn–Mg binary alloy (260 RPM, 4 h), which was subsequently compacted by selective laser melting (SLM). Obtained samples achieved a relative density of 97.8 % and an average grain size of 10 µm, suggesting that these materials will fail regarding mechanical properties and that the remelting during the SLM process completely transforms the fine structure of the powder. It is clear that up to date, only a limited amount of Zn-based alloys with similar or close grain sizes has been obtained. Material with grain size (590 ± 60 nm) has been prepared by the high-pressure torsion (HPT) for Zn–1Mg alloy [23], but this method is limited in the size of the affected regions. Significant work has been performed by Jarzębska and co-workers [24,25], who used hydrostatic extrusion to reach fine-grained microstructures with an average grain size slightly below 1 µm. Such behavior was attributed to the repetitive dynamic recrystallization during the process. An insight into the average grain sizes and related properties for selected Zn-based alloys with proximate composition to Zn–1Mg–0.5Sr is shown in Table 2.

### 3.2. Mechanical Properties

The mechanical properties of Zn–1Mg–0.5Sr alloy were evaluated based on HV1 measurements and compressive tests. The compressive stress–strain curves are presented in Figure 9, and the evaluated compressive yield strength (σ_CYS_) value and hardness measurement results are shown in Table 1. The compressive yield strength (σ_CYS_ = 346 MPa) and hardness (118 ± 2 HV1) are significantly increased compared to the pure zinc powder compacted by SPS. This improvement is related to the presence of extremely fine-grained microstructure, including the Mg_2_Zn_11_ particles. These hard intermetallic phases are presented in cubic crystallographic structure (Pm-3) and strengthen the material by blocking the movement of dislocation in the hexagonal lattice of Zn [30], but they can also lead to the loss of material plasticity [14]. However, the material continually deformed during the compression test (Figure 9) without reaching the fracture, even for a deformation of about 40 %, preserving excellent ductility. The hardness of the final product was mainly affected by the intermetallic phases with significantly higher hardness values than the surrounding matrix (330 HV for Mg_2_Zn_11_ [23], 356 HV for SrZn_13_ [31], 37 HV for Zn [23]). The contributions of grain boundaries (or grain size), solid solution, or intermetallic phases to the strengthening of the material were calculated previously for Zn–Mg alloys [6,22] and Zn-0.8Mg–0.2Sr [14]. It has been shown that the dominant effect is expected on the basis of the hindering movement of dislocations at grain boundaries and the Mg_2_Zn_11_/Zn matrix interface. On the contrary, the effect of solid solution strengthening may be considered negligible even for increased concentrations of Mg in solid solution [32]. 

Several studies were interested in zinc alloys with close composition. For example, Li et al. [12] studied the effect of processing on the mechanical properties of Zn–1.2Mg–1.1Sr alloy. They showed that the alloy reached the highest CYS value, equal to 375 MPa, after processing by extrusion, which is a very similar value to obtained results (327 MPa). The slightly higher σ_CYS_ is likely related to the higher concentration of Mg in the material and, therefore, higher content of Mg_2_Zn_11_, when the 0.3 wt.% of Mg difference cause approximately 26 MPa increase in σ_CYS_. Another contribution may be related to the presence of SrZn_13._ However_,_ their contribution is slightly reduced by prior distribution at grain boundaries or inside the Mg_2_Zn_11_ phase or eutectics in conventionally prepared materials. Materials prepared in this study further contained oxide shells, which are known to significantly affect the mechanical properties of materials prepared by SPS. The layer of oxides forms a network structure that runs through the entire alloy and causes a decrease in strength and ductility due to the weakening of the bonds between individual powder particles by the formation of a brittle oxide interface. Despite this shortcoming, the materials behave under compression loading superior to other Zn–Mg-based alloys with close composition (Table 2). This is attributed to the fact that the presented oxide network was partially disrupted due to the resistance heating at particle boundaries and possible local melting. These disruptions are essential to stop the easy propagation of cracks through the oxide shells thus preserving the material’s ductility.

## 4. Conclusions

The present study reveals the power of powder metallurgy processes in the synthesis of Zn-based materials with a homogeneous nano-grain microstructure. A combination of mechanical alloying and spark plasma sintering resulted in materials with the average grain size and intermetallic particle size of Zn–1Mg–0.5Sr alloy below 500 nm, which was observed for the first time for Zn-based bulk materials. Such microstructure refinement is related to the blocking of the grain boundaries by oxide particles formed in the material during mechanical alloying and preserving grain coarsening during compaction at increased temperatures. Microstructure refinement led to the compressive yield strength and hardness values of 86 HV1 a 327 MPa, respectively. Furthermore, the compacted materials retained high ductility in compression loading without any observed breakage. Therefore, the suggested technology brings new possibilities in increasing the performance of zinc-based alloys, particularly Zn–Mg–Sr and makes these materials candidates for bioabsorbable medical devices like augmentations or screws.

## Figures and Tables

**Figure 1 materials-15-08379-f001:**
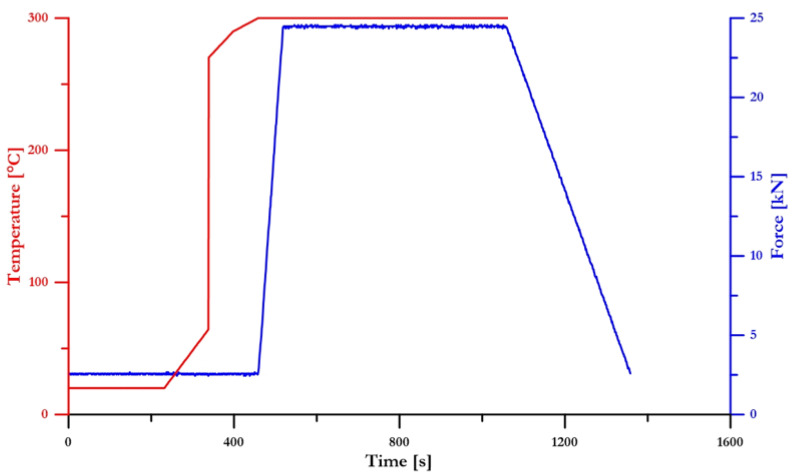
Time profiles of the SPS process conditions.

**Figure 2 materials-15-08379-f002:**
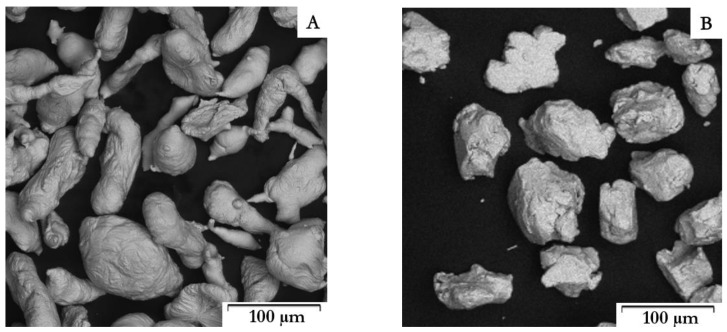
Powder particles: (**A**) Zn, (**B**) Zn–1Mg–0.5Sr.

**Figure 3 materials-15-08379-f003:**
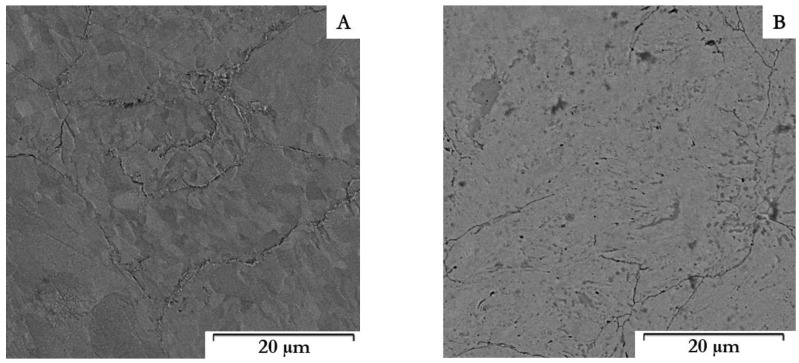
Microstructure of studied materials after SPS—scanning electron microscope (SEM): (**A**) Zn, (**B**) Zn–1Mg–0.5Sr.

**Figure 4 materials-15-08379-f004:**
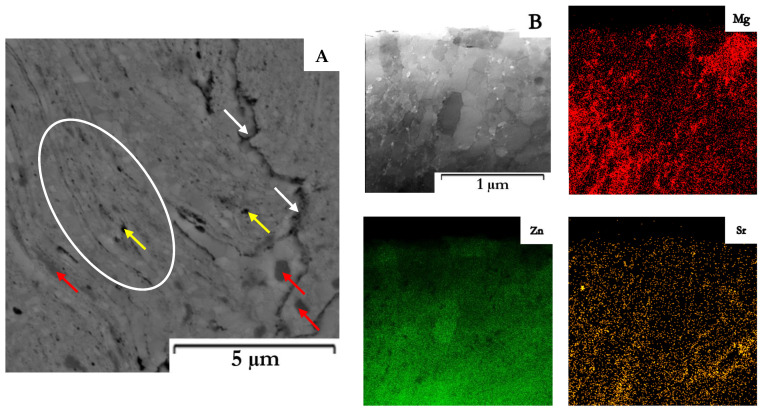
Microstructure of Zn–1Mg–0.5Sr alloy: (**A**) SEM, (**B**) Transmission electron microscope–Energy dispersive X-ray spectroscopy (TEM–EDS) map. Red arrows indicate Mg_2_Zn_11_ and yellow the residues of SrZn_13_ phases.

**Figure 5 materials-15-08379-f005:**
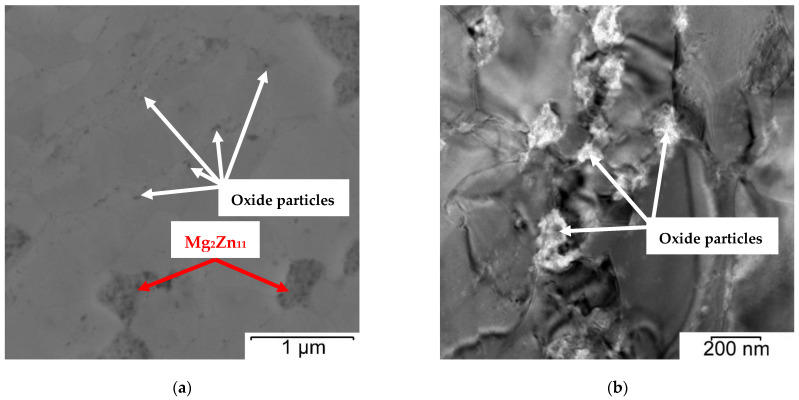
Microstructure of Zn–1Mg–0.5Sr with marked small oxide particles at grain boundaries and intermetallic phase Mg_2_Zn_11_: (**a**) overview (SEM), (**b**) detail (TEM).

**Figure 6 materials-15-08379-f006:**
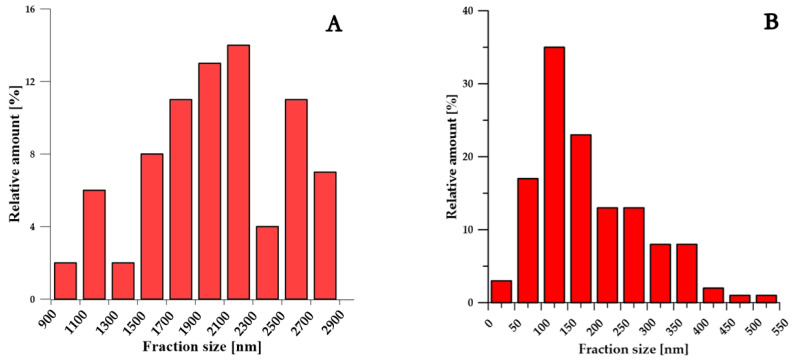
Grain size distribution: (**A**) Zn, (**B)** Zn–1Mg–0.5Sr.

**Figure 7 materials-15-08379-f007:**
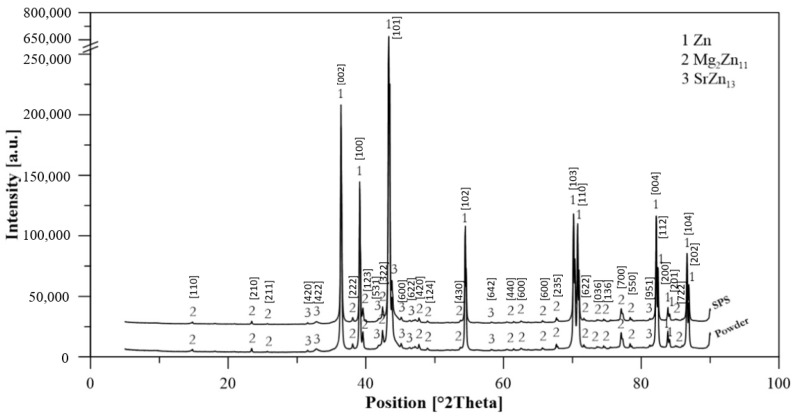
XRD diffractogram of mechanically alloyed Zn–1Mg–0.5Sr powder and compact.

**Figure 8 materials-15-08379-f008:**
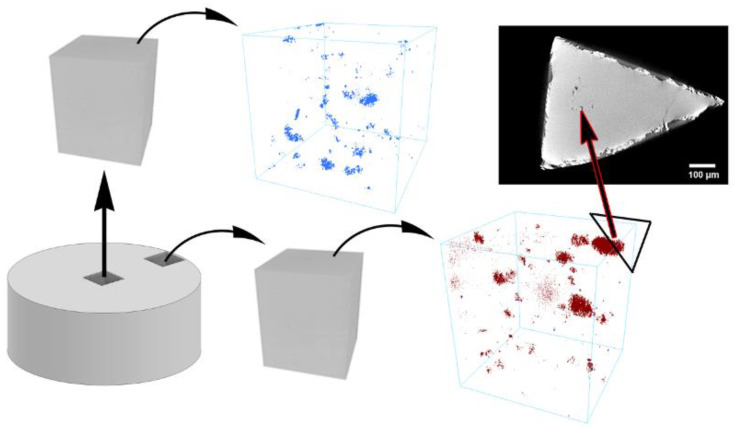
The porosity of the Zn–1Mg–0.5Sr from the center (blue pores) and the edge (red pores) of the billet with a 2D slice of cube corner acquired with a higher resolution to analyze the pore clusters.

**Figure 9 materials-15-08379-f009:**
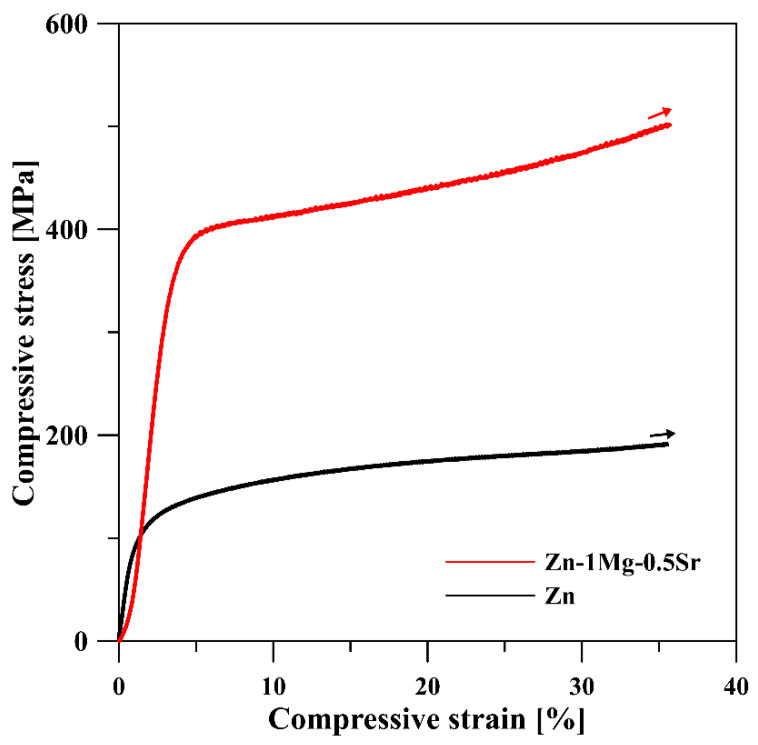
Compressive stress–strain curves of Zn–1Mg–0.5Sr alloy and Zn prepared by SPS method.

**Table 1 materials-15-08379-t001:** Chemical composition of Zn–1Mg–0.5Sr alloy (wt.%).

Sample Designation	Zn	Mg	Sr
Zn–1Mg–0.5Sr	98.7	0.9	0.4
Zn	100.0	-	-

**Table 2 materials-15-08379-t002:** Key characteristics of selected Zn-based alloys with close composition to Zn–1Mg–0.5Sr.

Composition	Synthesis	Grain Size [μm]	Hardness	CYS [MPa]	Ref.
**Zn**	Casting	500	38 ± 1	-	[12]
**Zn–0.03Mg**	ECAP (90°)	0.5–2.3	-	-	[26]
**Zn–1Mg**	HPT	0.6	250	-	[23]
**Zn–1Mg**	PM + Sintering	7.3	81 ± 5	245 ± 12	[3]
**Zn–1.6Mg**	Casting	35	82 ± 2	245 ± 12	[27]
**Zn–1.6Mg**	Extrusion	10	97 ± 3	292 ± 11	
**Zn–1.6Mg**	Melt spinning + extrusion	2	122 ± 3	382 ± 382	
**Zn–3Mg**	ECAP (120°)	2	186 ± 4	-	[28]
**Zn–0.6Mg–0.1Sr**	Casting	36	-	-	[17]
**Zn–0.6Mg–0.1Sr**	ECAP (150°)	3.6	-	-	
**Zn–0.8Mg–0.2Sr**	Extrusion	2–6	-	220 ± 6	[14]
**Zn-0.8Mg-0.2Sr**	ECAP	1–4	-	240 ± 10	[29]
**Zn–1Mg–1Sr**	Casting	10–50	85 ± 2	-	[12]
**Zn–1Mg–1Sr**	Rolling	10–50	-	383 ± 71	
**Zn–1Mg–1Sr**	Extrusion	10–50	92 ± 5	-	
**Zn**	MA + PM	-	38 ± 2	118 ± 2	This work
**Zn–1Mg–0.5Sr**	MA + PM	0.5	86 ± 2	327 ± 3	
**Zn–1Mg–0.5Sr**	Casting	-	82 ± 4	221 ± 6	Appendix A

CYS = compressive yield strength.

## Data Availability

Data available on request.

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
