# Peer review of "Ultrafine-Grained Zn–Mg–Sr Alloy Synthesized by Mechanical Alloying and Spark Plasma Sintering"

_materials, 2022, doi:10.3390/ma15238379_

Round 1

Reviewer 1 Report

1.  English language must be further improved to increase the quality of the manuscript.

2. The introduction part must be revised by adding a paragraph about SPS and MA methods.

3. Some of the sub-section title is repeated in few places, please assign different names to avoid the confusion to the readers.

4. As an expert of mechanical alloying, I doubt, 120 minutes is enough for the solid solution formation of the prepared alloy. As per the XRD and SEM, alloy has not completely undergone solid solution. I recommend to perform at least 10 hours of milling or else perform the optimization of milling parameters for better results.

5. SEM images of powders show the lumps of the Zn-Mg-Sr composition, not undergone refinement or plastic deformation. We cannot call it as an alloy. But after SPS, we can.

6. Figure 3 is confusing; author must mention that SEM of powders or SPS samples.

7. Figure 7, XRD shows no peak broadening or any shift of diffraction peaks compared between powder and the SPS samples. Authors must increase the milling time to obtain homogeneous and refined alloy.

Author Response

Dear sir or madam, thank you for your recommendations and suggestions. We uploaded answers to your questions in the attachment. 

Reviewer 2 Report

The paper is totally fine. It can be accepted after revision.

Some suggestions:

1.     The abstract and conclusion parts should be rewritten. Not only what have done, but also the key findings and novelty.

2.     Introduction: it is better to cite some recent papers about SPS of metals and alloys. E.g., Materials & Design, 2021, 210: 110108.

3.     How did the authors determine the sintering parameters? This needs clarify.

4.     Mechanical properties: it may be better to do tensile tests.

5.     More details about the experiments should be given. E.g., for XRD, the scanning speed, the target used, etc.

6.     Check the format of refs and polish the language.

Author Response

(The authors gave the same response as above.)

Reviewer 3 Report

Dear Authors,

The grammar need to be corrected from the native english speaker or by using the editorial service.

1. Check line no 15, s. In the presented work, we have

2. It is good to use base alloy or material in the place of pure Zn.

3. Reference no 16-17 is not releavant to this article.

4. The Introduction can be improved.

5. On what basis author chosen the temperature 100 deg c.

6. The microstructure is not clear and there is no clear cut information that how did the author measure the particle size from the image.

7. The individual powder morphology must be include.

8. A good TEM image can represent the precipitates well.

9. Fig 6 is not relevant and XRD graph must be validated with hkl values.

10. The conclusion must be re-write.

I request the author to refer the following paper and understand that how the precipitate look in the nm range ( https://www.sciencedirect.com/science/article/abs/pii/S0167577X2200074X)

Author Response

(The authors gave the same response as above.)

Round 2

Reviewer 1 Report

Dear Author, Thank you for considering my comments to improve the quality of the manuscript.

Author Response

Thank you

Reviewer 2 Report

Thanks for the nice revision.

Author Response

Thank you

Reviewer 3 Report

In general, the manuscript is not well organized and it is difficult to understand it after the review. The authors did not include any valid scientific discussion in the 2nd review. 80% of the suggestions provided by the reviewer is not addressed

Round 3

Reviewer 3 Report

To the authors:

I've read the comments, and I'm not convinced by the answers you've given.
Regarding this study, there are two points that require careful consideration.

For any comparison, it is preferable to compare the alloy to the base material, which is notably absent from the work. In addition, it is crucial to observe how your test circumstances compare to those stated in table 2. The authors must respond to the reviewers' comments in a constructive manner.

TEM studies are necessary to comprehend the precipitates in an alloy.
That is not discussed here.

Author Response

Dear Sir/Madam

Although we have some discrepancies in opinion on the presentation of further results and data, we tried to fulfil requirements and add required analyses to the paper or supplementary file.

Reviewer comment 1

For any comparison, it is preferable to compare the alloy to the base material, which is notably absent from the work. In addition, it is crucial to observe how your test circumstances compare to those stated in table 2. The authors must respond to the reviewers' comments in a constructive manner.

Regarding the required material for comparison, Zn-1Mg-0.5Sr alloy was cast and the microstructure and mechanical properties of such material were characterized. The data and description are shown in the supplementary file of the paper. Everything was performed by the same methodology as for the studied powder metallurgy products. We still think that these data are not exactly fit the intended paper structure, but according to the reviewer's suggestion, we would like to give the reader the possibility to have this comparison. Therefore we suggest having these data in a supplementary file like the attached one.

Reviewer comment 2

TEM studies are necessary to comprehend the precipitates in an alloy. That is not discussed here.

Regarding the additional TEM, we have performed further analyses and observed suggested oxide particles. These particles were analysed using EDS and data presented in the paper in the text marked by yellow on page 6. These analyses confirmed the presence of oxides in the microstructure and the content of both Mg and Zn in these oxides. Figure 5b has been added to the paper as the visualization of these oxides from TEM analyses.